# Wood-Inhabiting Nematode, *Bursaphelenchus ussuriensis* sp. n. (Nematoda: Aphelenchoididae) from David Elm, with Molecular Phylogeny of the Genus Based on Partial Mitochondrial Genomes

**DOI:** 10.3390/plants14010093

**Published:** 2024-12-31

**Authors:** Alexander Yu. Ryss, Sergio Álvarez-Ortega, Boris D. Efeykin, Ivan A. Kerchev, Kristina S. Polyanina, Anna I. Solovyeva, Sergei A. Subbotin

**Affiliations:** 1Laboratory for Parasitic Worms, Zoological Institute of the Russian Academy of Sciences, Universitetskaya Naberezhnaya 1, St. Petersburg 199034, Russia; alexander.yu.ryss@gmail.com (A.Y.R.); i.kristy17@mail.ru (K.S.P.); orcinuca@gmail.com (A.I.S.); 2Departamento de Biología y Geología, Física y Química Inorgánica, Universidad Rey Juan Carlos, Campus de Móstoles, 28933 Madrid, Spain; sergio.aortega@urjc.es; 3Instituto de Investigación en Cambio Global (IICG-URJC), Universidad Rey Juan Carlos, Tulipán s/n, 28933 Móstoles, Spain; 4Center of Parasitology of A.N. Severtsov Institute of Ecology and Evolution of the Russian Academy of Sciences, Leninskii Prospect 33, Moscow 117071, Russia; bocha19@yandex.ru; 5Institute of Monitoring of Climatic and Ecological Systems of the Siberian Branch of the Russian Academy of Sciences, Academichesky Avenue 10/3, Tomsk 634055, Russia; ivankerchev@gmail.com; 6Plant Pest Diagnostic Center, California Department of Food and Agriculture, 3294 Meadowview Road, Sacramento, CA 95832-1448, USA

**Keywords:** 28S rRNA gene, *Bursaphelenchus* spp., ITS rRNA gene, mitochondrial genomes, new species, phylogeny, Primorsky Territory

## Abstract

A new nematode species, *Bursaphelenchus ussuriensis* sp. n. is described in the bark beetle–elm tree association (*Scolytus jacobsoni* and *Ulmus davidiana* var. *japonica f. suberosa*) in the Asian Pacific region of Russia. The new species belongs to the *Hofmanni* group of *Bursaphelenchus* and is closest to *B. ulmophilus*. Its characteristics are as follows: lateral field with three incisures, body length 497–771 µm, post-uterine sac 3.6–5.4 times vulval body diam, 56 (39–66)% of vulva–anus distance, and spicule length 10.3 (9.5–12.5 µm). The new species differs from all species of the *Hofmanni* group in the closely situated P3 and P4 male caudal papillae and the GP5 small ‘glandpapillae’ pair on the butterfly-like papillae plate, in the set of P1, P2, P3, P4, GP5; vs. in all other species, the P4 papillae pair is absent in the pattern of P1, P2, P3, GP5. The phylogenetic position of *B. ussuriensis* sp. n. with other species of the *Hofmanni* group were reconstructed using the D2–D3 expansion segments of 28S and ITS rRNA gene sequence analysis. Sequences of twelve mitochondrial protein-coding genes of *B. cocophilus, B. fraudulentus, B. michalskii, B. ussurensis* sp. n., and *B. willibaldi* were obtained in this study. Phylogenetic relationships among eighteen *Bursaphelenchus* species based on the analysis of the mtDNA sequence dataset are provided and discussed. A modified diagnosis of the *Hofmanni* group is proposed.

## 1. Introduction

The genus *Bursaphelenchus* includes more than 140 species [1,2,3]. Nematodes of this genus are mostly fungivorous inhabiting freshly dead wood and utilizing various groups of insects as their phoretic vectors. Some *Bursaphelenchus* species are damaging pests of woody plants and two, *B. xylophilus* (Steiner & Buhrer, 1934) Nickle, 1970 and *B. cocophilus* (Cobb, 1919) Baujard, 1989, are considered economically important. *Bursaphelenchus xylophilus* causes the pine wilt disease of conifers with beetle vectors of *Monochamus* spp. (Cerambycidae) in North America, East Asia, and South Europe [4] while *B. cocophilus* is responsible for the devastating red ring disease of coconut palm (*Cocos nucifera* L.), oil palm (*Elaeis guineensis* Jacquin), and other palms, and is vectored by the palm weevil, *Rhynchophorus palmarum* L. in Central and South America [5,6]. In addition to these two species, ten other *Bursaphelenchus* species have been experimentally confirmed as weak to moderate plant pathogens [2].

Eighteen *Bursaphelenchus* species have been reported in Russia [7]. One of them, *B. ulmophilus* Ryss et al., 2015, was found in parks in St Petersburg [8] and is associated with Dutch elm disease (DED) of *Ulmus glabra*, caused by the European race of the fungus *Ophiostoma novo-ulmi* Brasier, 1991 [9,10]. The nematode is vectored by adults and larvae of the bark beetles *Scolytus multistriatus* Marsh. and *S. scolytus* Fabr. [8]. DED was first reported in the elms in the south of St Petersburg in 2002 [11,12] and is now widely distributed across the parks of the city and other regions [13].

In 2022–2023, during a nematology survey in the Ussuri State Natural Reserve Primorsky Territory, Russia, a new species of *Bursaphelenchus* was found in wood samples collected from dying elm trees, *Ulmus davidiana* var. *japonica f. suberosa* Nakai (Rehder) (Ulmaceae). The nematode was vectored by elm bark beetles, *Scolytus jacobsoni* Wood & Bright, 1992 (Curculionidae, Scolytinae). Elm trees showed symptoms of wilt disease, characterized by wilted crown and dark-colored ring in the cross-section of the wilted branches and galleries of beetle larvae and pupae in the trunk. Morphological and molecular analysis showed that this nematode species is closely related to *B. ulmophilus* and is described here as *B. ussuriensis* sp. n., a new member of the *Hofmanni* group and the *B. ratzeburgii* Rühm, 1956 species complex.

Present molecular identification of *Bursaphelenchus* species is based on the analysis of small subunit ribosomal RNA (18S rRNA), the D2–D3 expansion segments of large subunit ribosomal RNA (28S rRNA), and partial mitochondrial cytochrome oxidase subunit I (*cox1*) gene sequences. These genes have been used for the reconstruction of phylogenetic relationships within the genus [1,3,14,15,16,17,18]. Although the analysis of each of these genes and their combination showed robust relationships between some species and the presence of several distinct species groups, some relationships remain unresolved. More genomic information and markers will be necessary to reveal evolutionary trends in the genus. Although the mtDNA gene sequences included many phylogenetically informative sites, analysis of the *cox1* dataset did not resolve many interspecific relationships within *Bursaphelenchus* [3,14]. That may be explained by the use of short, amplified fragments (~627 bp) of this gene in most studies of this genus. Partial and whole mitochondrial genomes have been obtained for *B. xylophilus* and *B. mucronatus* and partial genomes for twelve *Bursaphelenchus* species: *B. fradulentus*, *B. tusciae*, *B. arthuri*, *B. borealis*, *B. fungivorus*, *B. gerberi*, *B. hofmanni*, *B. mucronatus*, *B. penai*, *B. platzeri*, *B. seani*, and *B. sexdentati*. Those genomes were used for the reconstruction of general nematode phylogeny by Gendron et al. [19]. Thus, the study of mitochondrial genomes of this nematode group remains insufficiently explored.

The main goals of the present study were to (i) describe a new *Bursaphelenchus* species associated with bark beetle, *Scolytus jacobsoni*, and elm trees from the Asian Pacific region of Russia; (ii) reconstruct phylogenetic relationships of the new species with other species from the *Hofmanni* group using rRNA gene sequences; and (iii) obtain partial mitochondrial genomes from new and several known *Bursaphelenchus* species to reconstruct phylogenetic relationships within the genus.

## 2. Results

### 2.1. Bursaphelenchus ussuriensis *sp. n.*

*Adults* (Figure 1, Figure 2, Figure 3, Figure 4 and Figure 5, Table 1) (https://zoobank.org/NomenclaturalActs/f84e1b12-4a02-4d1d-a41a-db49250668d5 (accessed on 29 December 2024)): body length *ca* 500–770 µm, slightly curved ventrally; stylet 10–13 µm long, its base slightly expanded, but without distinct knobs; cone 42–62% of its length (Figure 2C and Figure 4B). Cephalic annuli weakly visible under light microscopy and SEM. Nerve ring just posterior to the pharyngo–intestinal junction and median bulb. Secretory–excretory pore located from the posterior border of the nerve ring up to the level of the median bulb valve (Figure 2H and Figure 4C). Lateral field 3 µm wide, with two bands, appearing as three lines in the superficial view (Figure 5E,F).

*Male*: Male similar to female in structure of anterior end. Testis is 2/3 of the gonad length, at right subventral side of mid-intestine, anteriorly reflexed and with tightly packed polygonal spermatocytes, zone of spermatids distinct, consisting of two or three quartets of large separated cells, a zone of large light granulated ellipsoid immature sperm cells located posterior to spermatids, sperm gradually decreasing in size to dark mature sperm cells filling posterior 10–15% part of the sperm duct which has thick walls consisting of dark granular polygonal cells (Figure 2J). These cells presumably have a secretory function as the ellipsoid sperm cells 7 × 3.5 µm situated amongst the secretion granules. The tail strongly hooked like an umbrella handle, terminating with a mid-sized, membrane-like bursa with a central chord; its length 5–7.5 µm along the central line, 8–12 µm at borders, and 10 µm wide at base. Bursa spade-like with a truncate end with a straight posterior edge or sometimes slightly bifurcated at lateral borders (Figure 2D,F and Figure 5G,I). Male tail papillae in a complete set (P1–P4 and glandpapillae GP5). P1 unpaired immediately on the anterior cloacal lip, surrounded by a cuticular nest-like border; P2 paired, ventrolateral, at the same level as P1; P3 and P4 tightly contacted forming a duplet of papillae in lateral view; GP5 glandpapillae on the butterfly-like papillae plate, slightly posterior to P3 and P4 (Figure 2D,F,G and Figure 5G,H). Spicules stout, rostrum, and condylus well developed and separated (= *Hofmanni* group). The angle between lines along the capitulum (condylus-rostrum) and extending spicule end = −4.8–10.0°, point of intersection dorsal. Rostrum bluntly conical to rounded. Junction of rostrum and calomus rectangular to blunt. Condylus hemispherical to digitate; it is a continuation of the dorsal contour of spicules in most individuals, or sometimes slightly bent dorsally. Spicular tip (lateral view) with very small rounded cucullus, *ca* 1.0 µm diam, sometimes indistinct. Spicular lamina mid-point widened, bearing two ridges, as typical for the *Hofmanni* group, comprising one curved line along the dorsal lamina and a second straight line along the ventral lamina, both joining into the central line in the posterior conical part of the spicule. The dorsal spicular lamina smoothly and symmetrically curved (Figure 2D–G).

*Female*: Ovary well developed, not reaching pharyngeal gland lobe, situated on right subventral side of mid-intestine (Figure 4A). Oviduct straight and wide. Spermatheca oval, its length 2–2.5 times its diam., situated ventrally and to the left side of the posterior part of the oviduct, filled with ellipsoid to spherical cytoplasmic sperm 7 × 3.5 µm and small central nucleic part of 1 µm diam (Figure 4E). Spermatheca opening to oviduct just before oviduct opening to crustaformeria. Crustaformeria formed by spherical cells containing cytoplasmic granules, joining with the anterior uterus, walls of which consist of large flattened cells (Figure 3D,E). The vagina cuticular, slightly sloping anteriorly to the ventral body surface, vulval flap small in lateral view; from the ventral view vulva is a transverse slit with posteriorly curved margins, and an expanded smooth posterior lip; on the lip, the body annulation is interrupted (Figure 4D,E and Figure 5D). No vulval papillae or stripe-like striation of the vulval lips. Pair of two-celled well-sclerotized structures (vaginal framework) situated laterally on both sides of the vagina at the uterus/post-uterine sac junction, bearing a prong-like structure on the inner surface of the uterine wall (*vfr*, Figure 4D,E). Post-uterine sac (PUS) very wide, situated on the left subventral side, empty or filled with sperm cells, its end hemispherical, not differentiated, and devoid of rudimentary ovary (*pus*, Figure 4D,E). The ratio of PUS length to vulval body diam. = 3.6–5.4. PUS, forming 39–66% of vulva–anus distance. The tail reflexed, strongly hooked ventrally. The tail tip digitates to conical (Figure 4F and Figure 5E,F).

*Dauer juveniles* from elytrae of *Scolytus jacobsoni* were used as the inoculum for the cultivation in vitro. However, the dauer specimens were not fixed, photographed, or measured; thus, the description is based on the cultivated adult individuals only. However, the association with the vector, *S. jacobsoni* of the new species is established.

*Etymology:* The specific epithet is formed from the type locality of the new species, the Ussuri State Natural Reserve.

*Type habitat and locality*: Type materials were obtained from a culture on the fungus *B. cinerea* -PDA medium. The culture (code BULM-FE) was started from dauer individuals isolated from the elytra of beetle *Scolytus jacobsoni* obtained from a dying elm *Ulmus davidiana* var. *japonica* f. *suberosa* (Ulmaceae). The elm was characterized by symptoms of wilt disease (Dutch elm disease) with a wilted crown and dark-colored ring in the cross-section of the wilted branches; its trunk contained galleries of larvae and pupae of *S. jacobsoni*. Wood and insect materials were collected by I.A. Kerchev in Primorsky Krai, in the arboretum of the V.L. Komarov Mountain Taiga Station near the Ussuri State Natural Reserve on 2 August 2022, GIS coordinates: 43.689453 N, 132.156478 E. Sample codes in the CDFA collection were CD3963 and CD4139.

An additional sample (sample-2, culture code IK-3, CD4140) was started using inoculum of individuals extracted from beetle galleries in bark of the same host plant with the same beetle vector, collected in the neighborhood to the type locality on 13 August 2023, GIS coordinates: 43.693111 N, 132.146278 E.

*Type materials:* Type material obtained from 2-week-old cultures. A holotype male, 20 paratype females, and 20 paratype males are deposited in the Nematode Collection of the Zoological Institute RAS (UFK ZIN RAS), St. Petersburg, Russia. Four paratype males and four paratype females are also deposited in the Nematode Collection of Wageningen Agricultural University, The Netherlands, and four paratype males and paratype four females in the Nematode Collection of the University of California, Riverside, CA, USA. Additional paratypes (not less than 1000 individuals) propagated in cultures in vitro, fixed in hot TAF, and processed to anhydrous glycerine, are stored in the tube N TC-PPN-003 (tube glycerine collection of the UFK ZIN RAS); 15 slides containing 120 nematode individuals and the tube with *ca* 1000 of the TAF fixed and processed to glycerine nematodes of the location-2 (culture code IK-3) of the *B. ussuriensis* sp. n. are stored also in the UFK ZIN RAS.

*Diagnosis and relationships: Bursaphelenchus ussuriensis* sp. n. is characterized by lateral field with two bands (three incisures), spicule length 10.3 (9.5–12.5 µm) (arc, midline), relatively broad and short with small rounded condylus, straight or slightly bent dorsally, small distinct cucullus, spicule median surface with two ridges, dorsal lamina smoothly curved. Female: body length 0.50–0.77 mm; median bulb L/D ratio 1.4–1.6; spermatheca oval and oblong with L/D ratio 2–4; PUS 3.6–5.4 of vulval body diam. and 56 (39–66) % of vulva–anus distance usually filled with sperm; tail reflexed, strongly hooked ventrally, c’ value 4.5 (3.8–5.5). The tail tip digitates to the conical.

*Bursaphelenchus ussuriensis* sp. n. belongs to the *Hofmanni* group according to its position in molecular phylogeny and the morphological characteristics: lateral field with two bands (three incisures), spicules are relatively broad and short with condylus small, straight, or slightly bent dorsally, small cucullus present, spicule median surface with two ridges, dorsal lamina smoothly curved, not angular; insect vectors are the Scolytinae.

From all species of the *Hofmanni* group, the new species differs in closely situated P3 and P4 male caudal papillae and the GP5 small ‘glandpapillae’ pair on the butterfly-like papillae plate in the set of P1, P2, P3, P4, GP5, vs. in all other species of the *Hofmanni* group, the papilla-like P4 papillae are absent in a set of P1, P2, P3, GP5 (Appendix A).

Based on morphology, molecular and biological characteristics, *B. ussuriensis* sp. n. could be assigned to the *Bursaphelenchus ratzeburgii* species complex, which also includes *B. ratzeburgii*, *B. ulmophilus* and an undescribed *Bursaphelenchus* sp., named as *‘B. ophiostomae’* and recovered from larval galleries of elm bark beetles, *Scolytus* spp. in Poland.

*Bursaphelenchus ussuriensis* sp. n. differs from *B. ulmophilus* in the more oval median bulb with L/D ratio 1.4–1.6 vs. 1.2–1.3 in *B. ulmophilus*; PUS 3.6–5.4 vs. 2.6–3.4 times the vulval body diam. in *B. ulmophilus*; spicule length (arc, midline) 10.3 (9.5–12.5) vs. 16.1 (13.0–18.0) µm; c’ value 4.5 (3.8–5.5) vs. 3.4 (3.1–3.5) in *B. ulmophilus*; and the presence of the paired male caudal papillae P4 closely located to P3, while in *B. ulmophilus* the true P4 is absent and the GP5 pair is attributed as P4 in the original description, being the true glandpapillae, i.e., situated on the GP plate on mid-line at the border with the bursal flap ([8] p. 687). With regard to host–parasite relations, the new species is found in a similar *Scolytus-Ulmus* association; however, in the *Scolytus jacobsoni*—*Ulmus davidiana* var. *japonica* f. *suberosa* in the East Asian (Pacific) Russia, while the *B. ulmophilus* is collected in European Russia in the association of *Scolytus multistriatus* and *Sc. scolytus* in *Ulmus glabra* and *U. laevis* [8]. *Bursaphelenchus ussuriensis* sp. n. differs from *B. ratzeburgii* in the ratio of PUS to vulva–anus distance = 0.56 (0.39–0.66) vs. 0.29 (0.28–0.3) (calculated from figures in original species description), female tail tip reflexed and digitate vs. not reflexed and mucronate.

Among species of the *Hofmanni* group, the new species is close to *B*. *hofmanni* Braasch, 1998 in body and spicule shape. It differs from the latter in the shape of condylus (rounded, usually slightly offset from lamina border vs. angular and straight in *B. hofmanni*), the shape of rostrum (digitate or blunt vs. pointed in *B. hofmanni*, the ratio of PUS to vulva–anus distance 0.56 (0.39–0.66) vs. 0.3–0.5 in *B. hofmanni* [20].

*Bursaphelenchus ussuriensis* sp. n. differs from *B. anamurius* Akbulut, Braasch, Baysal, Brandstetter & Burgermeister, 2007 in the condylus well developed vs. completely reduced, vulval flap small but distinct vs. absent, tail tip reflexed vs. only slightly curved; from *B. corneolus* Massey, 1966 in spicule shape with condylus and rostrum of equal length vs. condylus very long and truncate and rostrum very short; from *B. curvicaudatus* Wang, Yu & Lin, 2005 in spicule length 10.3 (9.5–12.5) vs. 19 (17–22) μm, stylet length 9-13 vs. 14-17 μm, paired posterior male papillae (‘glandpapillae’) vs. unpaired single papilla, spicule condylus slightly reflexed dorsally vs. dorsal line of condylus being a straight continuation of the spicule dorsal lamina line; from *B*. *decraemerae* Wang, Gu, Munawar, Fang & Li, 2018 in PUS length 68–99 vs. 38–47 µm, PUS/V-anus distance ratio 0.56 (0.39–0.66) vs. 0.35 (0.30–0.39), condylus short, rounded vs. condylus distinctly elongated, P4 in *B. decraemerae* was described as ‘glandpapillae’ (homologous to GP5), in the complete set of caudal male papillae, while the papilla-like P4 is absent in *B.* decraemerae vs. present in the *B. ussuriensis* sp. n.

*Bursaphelenchus ussuriensis* sp. n. differs from *B. gerberi* Giblin-Davis, Kanzaki, Ye, Center & Thomas, 2006 in spicule shape with the line along the capitulum (condylus-rostrum) and line extending the spicule end crossing dorsally vs. ventrally (this character was used in [8,21,22,23], condylus and rostrum equally prominent vs. condylus very long and truncate and rostrum short; from *B. mazandaranense* Pedram, Pourjam, Ye, Atighi, Robbins & Ryss, 2011 in PUS length = 82 (68–99) vs. 45 (38–57) μm, and the ratio of PUS to VBD = 4.8 (3.6–5.4) vs. 2 (1.5–2.5); from *B. paracorneolus* Braasch, 2000 in its spicule shape with line along capitulum (condylus-rostrum) and line extending the spicule end crossing dorsally vs. ventrally; spicular lamina mid-point of *B. ussuriensis* sp. n. moderate in width and not mitten-shaped vs. excessively widened to mitten-shaped, condylus slightly flexed dorsally vs. condylus dorsal line appearing as a straight continuation of dorsal lamina, ratio of PUS to VBD = 4.8 (3.6–5.4) vs. 2.0 (1.7–2.4) (calculated from measurements in original species description); from *B. paraparvispicularis* Gu, Wang, Duan, Braasch, Burgermeister & Zheng, 2010 in the presence of a small cucullus at the spicule tip vs. lack of a distinct cucullus, presence vs. absence of a vulval flap, female tail relatively long with c’ = 4.5 (3.8–5.5) vs. 2.8 (2.4–3.2), and tail tip reflexed vs. only slightly ventrally curved; from *B. parapinasteri Wang & Zhang*, 2007 in its spicule shape which has the condylus slightly reflexed vs. perpendicular to the dorsal lamina of spicule, cucullus present vs. absent, bursa truncated to bifurcated vs. narrowly conical, female tail tip reflexed ventrally vs. slightly curved ventrally and almost straight; from *B. parvispicularis Kanzaki & Futai*, 2005 in its female tail tip reflexed vs. slightly curved, tail tip conical to digitate vs. cylindrically rounded, in spicule shape with line along the capitulum (condylus-rostrum) and line extending the spicule end crossing dorsally vs. ventrally (this character was used in: [21,22,23], the spicules are of moderate width with a prominent condylus flexed dorsally vs. broad spicules and compact straight condylus; from *B. pinasteri* Baujard, 1980 in its cucullus on spicule tip present vs. absent, and in spicule shape with line along capitulum (condylus-rostrum) and line extending the spicule end crossing dorsally vs. ventrally, female tail tip reflexed vs. straight; from *B. rufpennis* Kanzaki, Giblin-Davis, Cardoza, Ye, Raffa & Center, 2008 it differs in its spicule length (arc) 10.3 (9.5–12.5) vs. 12.5 (11.5–13.5) µm, condylus rounded short vs. elongated squared with posterior recurvature, bursa truncated to bifurcated vs. rounded, female tail tip hooked and narrow vs. tail cylindrical straight to slightly curved ventrally with rounded tip, c’ 4.5 (3.8–5.5) vs. 3.4 (3.0–3.9); from *B. sachsi* Rühm, 1956 in its spicule shape with line along capitulum (condylus-rostrum) and line extending the spicule end crossing dorsally vs. ventrally, cucullus present vs. absent, ratio of spicule length to its width posterior to rostrum = 2.5–4.0 vs. 5.1 (calculated from figures in original species description), female tail tip reflexed vs. slightly curved, and bursa spade-like to bifurcated vs. narrowly conical; from *B. scolyti* Massey, 1974 in its PUS/VBD 3.6–5.4 vs. 5–7, body length in female 554–771 vs. 840 µm, condylus short rounded vs. recurved, squared, cucullus small rounded vs. absent, glandpapillae GP5 in the caudal papilla set P1, P2, P3, P4, GP5 vs. small GP4 in the caudal papilla set P1, P2, P3, GP4 [24,25].

*Bursaphelenchus ussuriensis* sp. n. differs from *B. wuae* (Huang & Ye, 2006) Gu, Kanzaki & Tomalak, 2017 (with synonyms: =*B. osumiana* Kanzaki, Akiba, Kanetani, Tetsuka & Ikegame, 2014; =*B. yuyaoensis* Gu, He, Wang & Chen, 2014) in its spicule shape with the ratio of spicule length to its width posterior to rostrum = 2.5–4.0 vs. 1.9–2.1 (calculated from figures in original species description), condylus prominent vs. condylus small, not developed, cucullus well developed vs. cucullus small and indistinct, the ratio PUS/VBD = 4.8 (3.6–5.4) vs. 6.6 (5.4–7.2) (calculated from measurements in the original species description of *B. osumiana*); from *B. zealandicus* Zhao, Gu, Greenwood, Rogan, Ho & Taylor, 2024 in the presence of short vulval flap vs. lacking vulval flap, bursa well developed vs. oblong and narrow, condylus short rounded vs. condylus broad squared and recurved dorsally, spicule tip with small rounded cucullus vs. cucullus absent and spicule tip pointed, gland-papillae GP5 in the caudal papilla set P1, P2, P3, P4, GP5 vs. glandpapillae GP4 in the caudal papilla set P1, P2, P3, GP4, c’ in females 4.5 (3.8–5.5) vs. 3.6 (3.1–4.4).

*Bursaphelenchus xerokarterus* Rühm, 1956 is considered here *species inquirenda*; it was excluded from the comparison because its poor morphological description does not allow it to be classified into any of the species groups [26]. The species was an associate of Ulmaceae (*Ulmus foliacea* Gilib. and *Zelkova* sp.) and vectored by *S. scolytus* and *S. multistriatus* [27]. The species needs to be re-isolated and re-described [25].

*Molecular characterization of* B. ussuriensis sp. n.: the D2-D3 of 28S, ITS1-5.8S-ITS2 rRNA, and *cox1* gene sequences were obtained for this species from PCR amplicons. The whole length of 18S, ITS, and 28S rRNA genes were assembled from the WGS dataset. The phylogenetic position of *B. ussuriensis* sp. n. is within some representatives of the *Hofmanni* group based on the analysis of D2–D3 of 28S and ITS rRNA gene sequences. Sequences of *B. ussuriensis* sp. n. clustered with those of *B. ulmophilus* and *Bursaphelenchus* sp. in the phylogenetic trees with high statistical support values (PP = 100), and relationships within these species were not well resolved (Figure 6).

Two new D2-D3 of 28S rRNA gene sequences obtained from two samples of *B. ussuriensis* sp. n. (CD3963, CD4170) were identical. The D2–D3 of 28 rRNA gene sequence alignment included 27 sequences of 18 *Bursaphelenchus* species and was 786 bp in length. The D2–D3 of 28 rRNA gene sequences of *B. ussuriensis* sp. n. differed from those of *B. ulmophilus* in 1.2% (9 bp), *Bursaphelenchus* sp. in 1.6% (11 bp), and *B. ratzeburgii* in 1.1% (8 bp). The difference between sequences of *B. corneolus* and *B. curvicaudatus* was 1.3–1.9% (10–13 bp).

One new ITS rRNA gene sequence was obtained for *B. ussuriensis* sp. n. The ITS rRNA gene sequence alignment included 27 sequences of 18 *Bursaphelenchus* species and was 1329 bp in length. The ITS rRNA gene sequence of *B. ussuriensis* sp. n. differed from those of *B. ulmophilus* in 4.4% (37 bp), *Bursaphelenchus* sp. in 6.2% (52 bp), and *B. ratzeburgii* in 4.0–4.3% (34–36 bp). The difference between sequences of *B. corneolus* and *B. curvicaudatus* was 4.3-5.0% (49–55 bp).

### 2.2. Mitochondrial Genes and Molecular Phylogeny of the Genus Bursaphelenchus

Sequences of twelve mitochondrial protein-coding genes: *cox1*, *cox2*, *cox3*, *atp6*, *cytb*, *nad1*, *nad2*, *nad3*, *nad4*, *nad4L*, *nad5*, and *nad6* for five *Bursaphelenchus* species: *B. cocophilus, B. fraudulentus, B. michalskii, B. ussurensis* sp. n., and *B. willibaldi* were obtained in this study (Appendix A). New sequences of two populations of *B. cocophilus* were included in the analysis. Lengths of concatenated mitochondrial gene nucleotide sequences for these species were: *B. cocophilus*—10,302 bp and 10,304 bp, *B. fraudulentus*—10,163 bp, *B. michalskii*—10,309 bp, *B. ussurensis* sp. n.—10,162 bp, and *B. willibaldi*—10,302 bp. Lengths of mitochondrial gene sequences for twelve *Bursaphelenchus* species obtained by Gendron et al. [19] after extraction from the database and our verification were as follows: *B. arthuri*—3933bp, *B. borealis*—6565 bp, *B. fradulentus*—5898 bp, *B. fungivorus*—5744 bp, *B. gerberi*—5284 bp, *B. hofmanni*—7726 bp, *B. mucronatus*—9421 bp, *B. penai*—7284 bp, *B. platzeri*—9019 bp, *B. seani*—8726 bp, *B. sexdentati*—6607 bp, and *B. tusciae*—6468 bp. Intraspecific nucleotide sequence variations between two populations of *B. cocophilus* and *B. fradulentus* were 0.5% and 1.7%, respectively, and two subspecies of *B. mucronatus mucronatus* and *B. mucronatus kolymensis* were 8.9%.

Nucleotide and amino acid alignments included 18 *Bursaphelenchus* species and two outgroup species and were 10,471 bp and 3491 aa in length, respectively (Appendix A). Phylogenetic relationships between species of the genus *Bursaphelenchus* are given in Figure 7. These species formed seven species groups. *Bursaphelenchus ussuriensis* sp. n. clustered with *B. hofmanni* with a high statistical support value (PP = 100) on nucleotide and amino acid trees.

## 3. Discussion

In this study, we describe a new species, *B. ussurensis* sp. n., from the *Hofmanni* group, a group which includes species associated with coniferous and deciduous woody plants and vectored by representatives of the family Curculionidae, subfamily Scolytinae [8]. It is possible that the new species is associated with DED and a study on fungi associated with this nematode and insect vector is underway.

### Modified Diagnosis of the Hofmanni Group

Macroclade C within the molecular phylogenetic cladogram [1]. Lateral field with three incisures (two bands). Male caudal papillae pattern includes the papilla-like papillae: unpaired P1 anterior to cloacal lip, paired P2 located ventro-lateral at the same level as P1; paired P3 (or converged P3+P4) and a pair of small pore-like papillae P5 (‘glandpapillae’) at the base of the bursa; a pair of papillae P4 is absent or strongly converged with P3, so under the light microscope, they appear to be P3+P4 fused together. Spicules are relatively broad and short with a small straight or slightly dorsally bent condylus, cucullus present, and two ridges along the median line of the spicule lamina. Life cycle trixenous with three associates: the beetles of subfamily Scolytinae are vectors of the transmissive juveniles (dauers), hosts of the propagative generation are fungi as well as the coniferous and deciduous woody plants.

*Hoffmanni* group of the genus *Bursaphelenchus* with the list of 22 valid species:*B. anamurius* Akbulut, Braasch, Baysal, Brandstetter & Burgermeister, 2007*B. corneolus* Massey, 1966*B. curvicaudatus* Wang, Yu & Lin, 2005*B. decraemerae* Wang, Gu, Munawar, Fang & Li, 2018*B. gerberi* Giblin-Davis, Kanzaki, Ye, Center& Thomas, 2006*B. glaucae* Kanzaki & Fujimori, 2024*B*. *hofmanni* Braasch, 1998*B. mazandaranense* Pedram, Pourjam, Ye, Atighi, Robbins & Ryss, 2011*B. moensi* Wang, Munawar, Gu, Fang, Wang & Li, 2018b*B. paracorneolus* Braasch, 2000*B. paraparvispicularis* Gu, Wang, Duan, Braasch, Burgermeister & Zheng, 2010*B. parapinasteri Wang & Zhang*, 2007*B. parvispicularis Kanzaki & Futai*, 2005*B. pinasteri* Baujard, 1980*B. ratzeburgii* Rühm, 1956*B. rufpennis* Kanzaki, Giblin-Davis, Cardoza, Ye, Raffa & Center, 2008*B. sachsi* Rühm, 1956*B. scolyti* Massey, 1974*B. ulmophilus* Ryss, Polyanina, Popovichev & Subbotin, 2015*B. ussuriensis* sp. n.*B. wuae* (Huang & Ye, 2006) Gu, Kanzaki & Tomalak, 2017 (= *Bursaphelenchus yuyaoensis* Gu, He, Wang & Chen, 2014; = *Bursaphelenchus osumiana* Kanzaki, Akiba, Kanetani, Tetsuka & Ikegame, 2014)*B. zealandicus* Zhao, Gu, Greenwood, Rogan, Ho & Taylor, 2024


*Species inquirenda*


*Bursaphelenchus xerokarterus* Rühm, 1956

*Bursaphelenchus ussurensis* sp. n. is the fourth valid species of the *B. ratzeburgii* species complex proposed in our study. The complex consists of *B. ratzeburgii, B. scolyti,* and *B. ulmophilus. Bursaphelenchus ratzeburgii* is associated with *Scolytus ratzeburgii* Janson in Betulaceae trees, i.e., *Betula verrucosa* Ehrh. and *Betula* sp. in Germany and Georgia [23,27,28]. *Bursaphelenchus scolyti* collected from galleries of *Scolytus multistriatus* in the bark of the Dutch elm, *Ulmus americana* L., at Fort Collins, Colorado, USA is not yet molecularly characterized [24]. This species is typologically similar to *B. ratzeburgii* and *B. xerokarterus* based on female tail morphology: smoothly tapered and curved ventrally [25]. *Bursaphelenchus ulmophilus* was found associated with Dutch elm disease of *Ulmus glabra* in parks of St Petersburg, Russia, and is vectored by the bark beetles *Scolytus multistriatus* and *S. scolytus* [29]. Another species, *B. xerokarterus,* probably also belongs to this species complex; however, it is considered *species inquirenda* by Braasch et al. [26] and in this study. This species is associated with *S. scolytus* and *S. multistriatus* in several broad-leaved tree species: *Ulmus campestris* L., *U. pedunculata* Foug., *U. foliacea*, *Zelkova* sp. (Ulmaceae), *Carpinus caucasica* Grossh. (Betulaceae), *Juglans* sp. (Juglandaceae), and *Populus nigra* L. (Salicaceae) in Germany and Georgia [27,28]. An undescribed species, temporarily named *Bursaphelenchus ‘ophiostomae‘,* was found in larval galleries of elm bark beetles, *Scolytus scolytus* and *S. multistriatus* (Coleoptera: Curculionidae) in Poland. The importance of the study of the *B. ratzeburgii* species complex is justified by the roles and association of these nematodes with the development and distribution of the Dutch elm disease.

It is important to note that two other species are also associated with *Ulmus* spp. but that they belong to the *Eggersi* group: *B. michalskii* transmitted by *Scolytus scolytus* and found in Dutch elm disease-affected *Ulmus laevis* Pall. [30] and *B. laciniatae* isolated from the bark of dead *U. laciniata*, which harbored *S. esuriens* galleries, collected in Hokkaido, Japan [17].

Description of the new species, *Bursaphelenchus ussuriensis* sp. n., is based on an integrated approach combining morphological and molecular analysis. Nucleotide differences in the ITS-rRNA gene sequence of *B. ussuriensis* sp. n. from other representatives of the *B. ratzeburgii* complex are similar to those between *B. corneolus* and *B. curvicaudatus. Bursaphelenchus corneolus* was originally detected in ponderosa pine in New Mexico, USA [31] and sequences of this species were obtained from specimens identified under this name from packaging wood imported from Taiwan to China [32]. *Bursaphelenchus curvicaudatus* was isolated from conifer wood in cargo ships from Mexico [33]. Braasch et al. [32] did not accept the validity of *Bursaphelenchus curvicaudatus* and proposed subspecies assignments of *B. corneolus* with samples from Taiwan as *B. corneolus taiwanensis* and from Mexico as *B. corneolus corneolus* (syn. *B. curvicaudatus*). In this study, we do not recognize the subspecies categories of *B. corneolus* due to substantial morphological and molecular differences between the described species: *B. curvicaudatus* and *B. corneolus*. However, we agree with Kanzaki et al. [34] that detailed comparisons of molecular sequences and morphological characters are still necessary between the Taiwanese population of *B. corneolus* and the original USA populations before accepting a definitive conclusion. It is noteworthy that the only presently accepted subspecies in the genus *Bursaphelenchus* are *B. mucronatus mucronatus* and *B. mucronatus kolymensis* [35]. Nucleotide differences in the ITS-rRNA gene sequences between these European and Asian types of *B. mucronatus* are only 2.0–3.0% [35,36], which are smaller than the differences observed between species in the *B. ratzeburgii* complex and between *B. corneolus* and *B. curvicaudatus*.

The complete and basic for genus *Bursaphelenchus* pattern of male caudal papillae includes papilla-like P1, P2, P3, P4, and small glandpapillae GP5; the P1 is unpaired and other papillae are paired. In the *Hofmanni* group, the true P4 pair is absent, and the pattern is as follows: P1, P2, P3, GP5; and a small glandpapillae pair GP5 [1]. The converged tandem of papillae P3+P4 in *B. ussuriensis* sp. n. is a new characteristic now included in the modified diagnosis of the *Hofmanni* group. This new characteristic suggests the need for special attention to morphological re-examination of this group. Papillae P3 and P4 are located so close together that under the light microscope, they may appear as one pair of papillae rather than two independent pairs. Another important result of this study is the phylogenetic similarity of the new species to the already known species *B. ulmophilus*, which has phylogenetically closely related hosts and vectors but is found in different parts of Eurasia. It seems highly likely that there has been co-evolution of the ancestor of two close species with their associates in different geographical areas. A special study of the phylogeography of the stages of the co-speciation for the species pair is desirable. An additional physiological difference between two sibling species was revealed by in vitro experiments with the wood decay fungus *Bjerkandera adusta* (Willd.) P. Karst.: *B. ulmophilus* propagated successfully on the *B. adusta* culture, while *B. ussuriensis* sp. n. could not feed or propagate and eventually died [29].

In this study, we presented the phylogeny of *Bursaphelenchus* based on the analysis of twelve mitochondrial protein-coding genes for 18 species, which represent seven from eleven known species groups from this genus according to Ryss and Subbotin [1]. Our dataset provided improved resolution of interspecies and intergroup relationships compared to analyses that rely solely on partial sequences of the *cox1* gene [3,14]. The mitochondrial phylogeny also revealed the presence of the *Cocophilus* group containing only *B. cocophilus* and *B. platzeri*, which was previously proposed by Braasch et al. [26] but was not clearly supported using the ribosomal RNA gene-based phylogenies. These two species shared unique morphological features of the partly fused ventral limbs of the spicules, female tail shape, and phoresy with non-scolytid beetles [37].

With the description of this new species, the total number of *Bursaphelenchus* species reported from Russia increased to nineteen [7]; it is also the second species described from the Russian Far East. Further surveys in East Asian forests are still needed to determine species diversity and to understand biogeography and mechanisms of dispersal of this nematode group. The number of *Bursaphelenchus* species continues to increase and more mtDNA genes of these species will be necessary to understand the grouping and phylogeny of the genus.

## 4. Materials and Methods

### 4.1. Nematode Samples and Cultures

The wood samples with nematodes and beetles *Scolytus jacobsoni* were obtained from a dying elm *Ulmus davidiana* var. *japonica* f. *suberosa* (Ulmaceae) with symptoms of wilt disease near the Ussuri State Natural Reserve in Primorsky Territory, Russia. Nematode cultures were established with dauer individuals isolated from the beetle elytrae and from nematodes extracted from the bark using a modification of the Baermann funnel technique [38]. Cultures of the fungus *Botrytis cinerea* were maintained on 2% potato dextrose agar (PDA) prepared from fresh potatoes and agar [39]. Cultures were grown for one week at room temperature, until the white mycelium covered the agar surface, before nematode inoculation. To maintain nematode isolates in vitro, twenty nematode specimens (females, males, and juveniles) were inoculated into the mycelium. The nematodes increased to thousands of individuals per Petri dish in 8–16 days. Nematodes collected from two locations with sample codes CD3963, CD4139 (first location), and CD4170 (second location) were used for molecular study.

Cultures of four nematode species, *B. ussuriensis* sp. n., *B. fraudulentus, B. michalskii,* and *B. willibaldi* maintained as described above, and specimens of *B. cocophilus* fixed in ethanol were used for DNA extraction and preparation for whole genome sequencing (WGS). *Bursaplenchus cocophilus* was collected in Mexico [40], *B. fraudulentus* from Novosibirsk, Russia, *B. willibaldi* from Nizhny Novgorod, and *B. michalskii* from Dagestan, Russia [7] (Appendix A).

### 4.2. Light (LM) and Scanning Electron Microscopy (SEM) Studies

Nematodes were fixed in hot TAF (triethanolamine formalin) and slides were prepared by the express technique [41]. All slide-mounted nematodes were measured and photographed under an automated Leica DM5000 B microscope with differential interference contrast (DIC) and a Leica DFC320 (R2) digital camera with Leica DFC Twain Software (https://www.leica-microsystems.com/products/microscope-cameras/) (accessed on 30 December 2024) for PC and Leica IM50 Image Manager for PC. Illustrations were made using a camera lucida and a series of photographs. All measurements were analyzed using the software ImageJ 1.48v (National Institute of Health, USA, (https://imagej.net/ij (accessed on 30 December 2024)). Following their examination and identification, a few specimens were preserved in glycerin for observation under SEM using the protocol of Álvarez-Ortega and Peña-Santiago [42]. The nematodes were hydrated in distilled water, dehydrated in an ethanol and acetone gradient series, critical point-dried, coated with gold, and observed with a Nova NanoSEM 230 microscope (SEC Co. Ltd., Suwon, Republic of Korea).

### 4.3. DNA Extraction, PCR, Genome Amplification and Sequencing

Genomic DNA of *B. ussuriensis* sp. n. for PCR was obtained from several pooled nematodes using proteinase K. PCR and sequencing protocols were as described by Subbotin [43]. Several primer sets were used in this study. D2A (5′-ACA AGT ACC GTG AGG GAA AGT TG-3′) and D3B (5′-TCG GAA GGA ACC AGC TAC TA-3′) primers were used for amplification of the D2-D3 expansion segments of 28S rRNA gene; F194 (5′-CGT AAC AAG GTA GCT GTA G -3′) and AB28 (5′-ATA TGC TTA AGT TCA GCG GGT-3′) primers were used for amplification of the ITS rRNA gene; JB3 (5′-TTT TTT GGG CAT CCT GAG GTT TAT-3′) and JB5 (5′-AGC ACC TAA ACT TAA AAC ATA ATG AAA ATG-3′) were used for a for amplification of the partial *cox1* gene. Sanger sequencing of amplicons was conducted by Azenta (South San Francisco, CA, USA).

Genomic DNA of *B. ussuriensis* sp. n., *B. willibaldi*, *B. michalskii,* and *B. fradulentus* for whole genome sequencing was extracted with CTAB (cetyltrimethylammonium bromide) detergent according to the published protocol with modifications [44]. A volume of 50 µL of fresh nematodes from cultures was homogenized with CTAB buffer containing 2% of CTAB (Sigma-Aldrich, Inc., St. Louis, MO, USA), 100 mM TrisHCl ph 8 (Sigma-Aldrich, Inc., St. Louis, MO, USA), 20 mM EDTA (Helicon, Moscow, Russia), 1.4 M NaCl, 80 µg/mL of Proteinase K (Evrogen, Moscow, Russia), and 0.2% of β-mercaptoethanol (Sigma-Aldrich, Inc., St. Louis, MO, USA). Samples were incubated at 60 °C for 30 min; then, DNA was extracted with an equal volume of chloroform-isoamylalcohol (24:1) by centrifugation at 10,000 g and precipitated with isopropanol. DNA pellets were washed with 96% ethanol twice, dried, and dissolved in TE buffer (10 mM Tris HCl, 1 mM EDTA, pH 8). Next, 100 µg/mL of RNAse A (Thermo Fisher Scientific, Waltham, MA, USA) was added and samples were incubated at room temperature for 30 min. DNA was purified with KAPA Pure Beads (KAPA Biosystems, Wilmington, USA) according to the manufacturer’s protocol. DNA of *B. willibaldi* and *B. michalskii* was submitted for genome sequencing.

Genomic DNA of *B. cocophilus* was obtained from several pooled nematodes using proteinase K. DNA of *B. ussuriensis* sp. n., *B. fradulentus,* and *B. cocophilus* was enriched using the whole genome amplification (WGA) approach. WGA was performed using an Illustra GenomiPhi V2 DNA amplification kit (Cytiva, Marlborough, MA, USA) following the manufacturer’s instructions. The products were purified using a QIAquick PCR purification kit (Qiagen, Germantown, MD, USA) and submitted for genome sequencing, as detailed below.

### 4.4. Genome Sequencing and Mitochondrial Genome Assembly

DNA sample of *B. cocophilus* was submitted for library construction and WGS to Novogene Co., Ltd., (Sacramento, CA, USA). The DNA library was prepared using a NEBNext Ultra II DNA library prep kit from Illumina (New England Biolabs, Inc., Ipswich, MA, USA) following the manufacturer’s recommendations and sequenced on an Illumina NovaSeq 6000 instrument. DNA samples of *B. willibaldi* and *B. michalskii* were submitted to the RTSF Genomics Core at Michigan State University (East Lansing, MI, USA). DNA libraries were prepared using the Illumina TruSeq Nano DNA Library Preparation Kit, following manufacturers’ recommendations. The sequencing was performed on an Illumina NovaSeq 6000 instrument (San Diego, CA, USA). DNA samples of *B. fradulentus* and *B. ussuriensis* sp. n. were submitted to the Dante Lab (New York, NY, USA). DNA libraries were prepared using the Vazyme WGS Library prep kit and sequenced on the Illumina Novaseq X platform. The raw sequence reads were assembled de novo in contigs using SPAdes v3.15.4 [45]. Mitochondrial genome assembly was carried out using the algorithm NOVOPlasty v. 4.3.3 [46] or mapping on reference mitochondrial *Bursaphelenchus* genomes, *B. xylophilus* (NC_023208) and *B. mucronatus* (NC_021120) using Geneious 9.5 (https://www.geneious.com (accessed on 29 December 2024)). Mitochondrial genes were annotated using MITOS2 at Galaxy Version 2.1.9 (https://usegalaxy.org (accessed on 29 December, 2024)).

### 4.5. Phylogenetic and Sequence Analysis

The newly obtained D2–D3 expansion segments of 28S rRNA and ITS rRNA gene sequences of *B. ussuriensis* sp. n. were aligned with corresponding gene sequences of species of the *Hofmanni* published in the GenBank [3,8,14,32,47,48,49,50]. Outgroup taxa for each dataset were chosen according to previously published data [8]. Sequences of mtDNA genes for twelve *Bursaphelenchus* species: *B. fradulentus*, *B. tusciae*, *B. arthuri*, *B. borealis*, *B. fungivorus*, *B. gerberi*, *B. hofmanni*, *B. mucronatus*, *B. penai*, *B. platzeri*, *B. seani*, and *B. sexdentati* obtained by Gendron et al. [19] were downloaded from https://github.com/emsgendron/Nematode-Comparative-Mitochondrial-Genomics (accessed on 29 December 2024) and verified. Sequences were aligned using ClustalX [51]. ClustalX was run with default (gap opening—15.0; gap extension—6.66) parameters for the 28S rRNA and whole mtDNA gene datasets and modified (gap opening—5.0; gap extension—3.0) parameters for the ITS rRNA gene dataset. The nucleotide sequence alignments were analyzed under the GTR + I + G model and the amino acid sequence alignment under the MTMAM model with Bayesian inference (BI) using MrBayes 3.2.7 [52]. BI analysis for each dataset was initiated with a random starting tree and was run with four chains for 1.0 × 10^6^ generations for nucleotide alignments and 1.0 × 10^5^ for amino acid alignment. The Markov chains were sampled at intervals of 100 generations. Two runs were performed for each analysis. Posterior probabilities (PP) are given when they equal or exceed 70%. Pairwise divergences between taxa were computed with PAUP* 4b10 [53] as absolute distance values and as percentage mean distance values based on the entire alignment, with adjustment for missing data.

## Figures and Tables

**Figure 1 plants-14-00093-f001:**
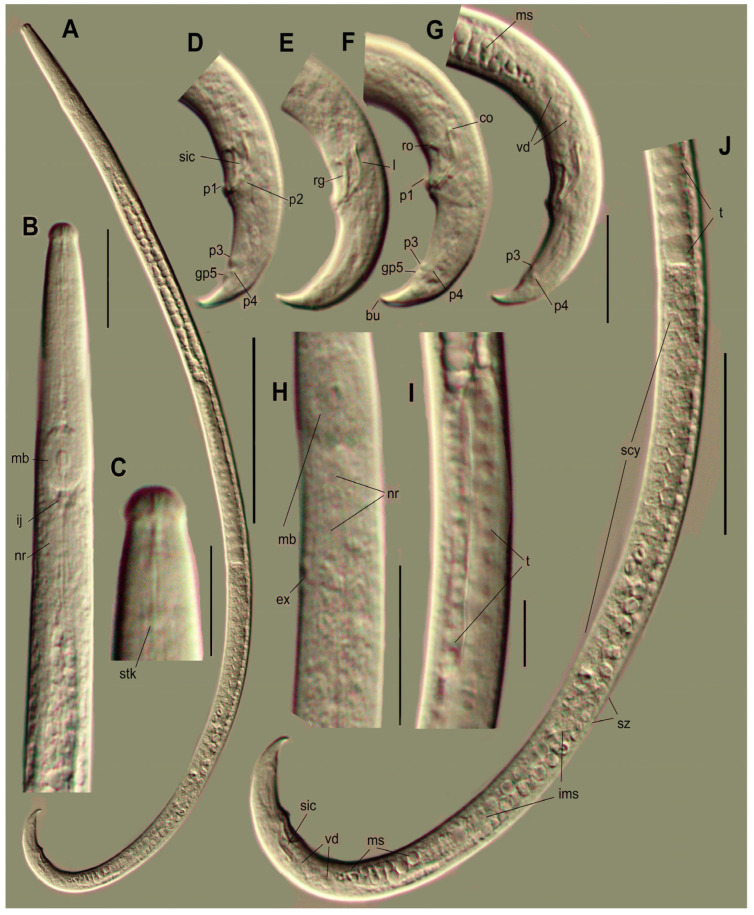
*Bursaphelenchus ussuriensis* sp. n. Male. Light microscopic photographs. (**A**)—total body outline. (**B**)—anterior part of the body. (**C**)—lip region, (**D**–**G**)—tails and spicules. (**H**)—excretory pore position (arrow). (**I**)–flexure of testis. (**J**)—posterior part of the body. Abbreviations: bu—bursal flap, co—condylus, ex—secretory–excretory pore, ij—pharyngo–intestinal junction, ims-immature sperm, l—lamina, mb—median bulb, ms—mature sperm, nr—nerve ring, p1, p2, p3, p4, gp5—male caudal papilla, rg—ridge along ventral velum of spicule, ro—rostrum, scy-spermatocytes, sic—spicule, stk—stylet knobs, sz—zone of spermatids, t—testis, vd—vas deferens. Scale: 100 µm for (**A**); 50 µm for (**J**); 10 µm for (**C,I**); 20 µm for rest.

**Figure 2 plants-14-00093-f002:**
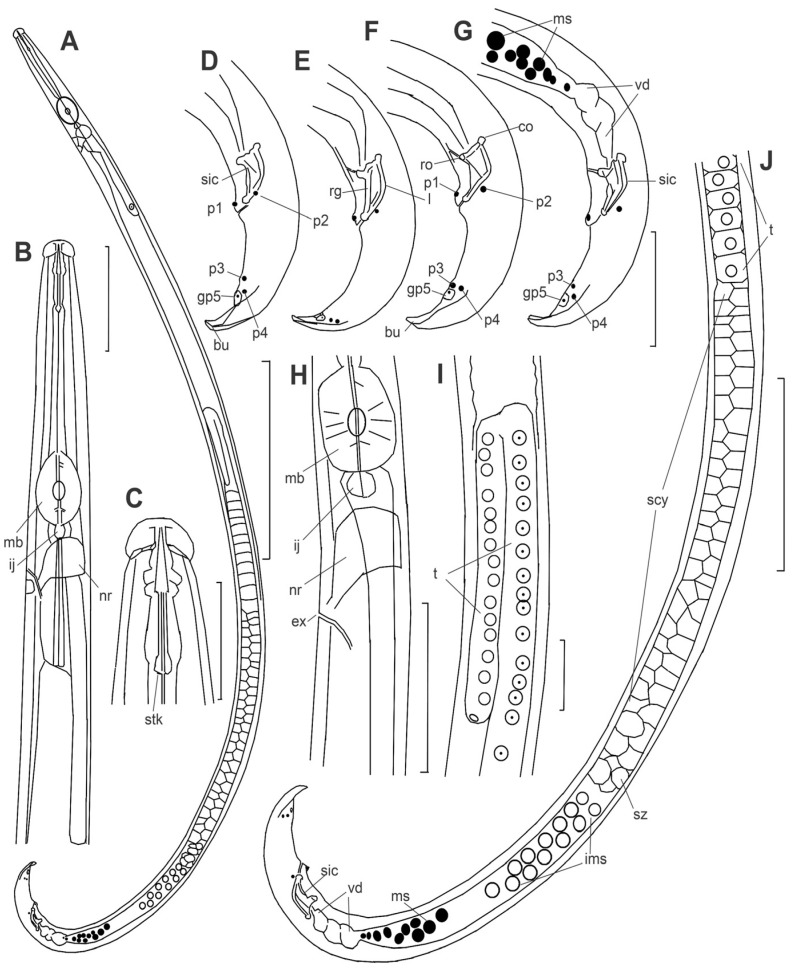
*Bursaphelenchus ussuriensis* sp. n. Drawings. Male. (**A**)—total body outline. (**B**)—anterior part of the body. (**C**)—lip region, (**D**–**G**)—tails and spicules. (**H**)—excretory pore position (arrow). (**I**)—flexure of testis. (**J**)—posterior part of the body. Abbreviations: bu—bursal flap, co—condylus, ex—secretory–excretory pore, ij—pharyngo–intestinal junction, ims-immature sperm, l—lamina, mb —median bulb, ms -mature sperm, nr—nerve ring, p1, p2, p3, p4, gp5—male caudal papilla, rg—ridge along ventral velum of spicule, ro—rostrum, scy-spermatocytes, sic—spicule, stk—stylet knobs, sz—zone of spermatids, t—testis, vd—vas deferens. Scale: 100 µm for (**A**); 50 µm for (**J**); 10 µm for (**C**,**I**); 20 µm for rest.

**Figure 3 plants-14-00093-f003:**
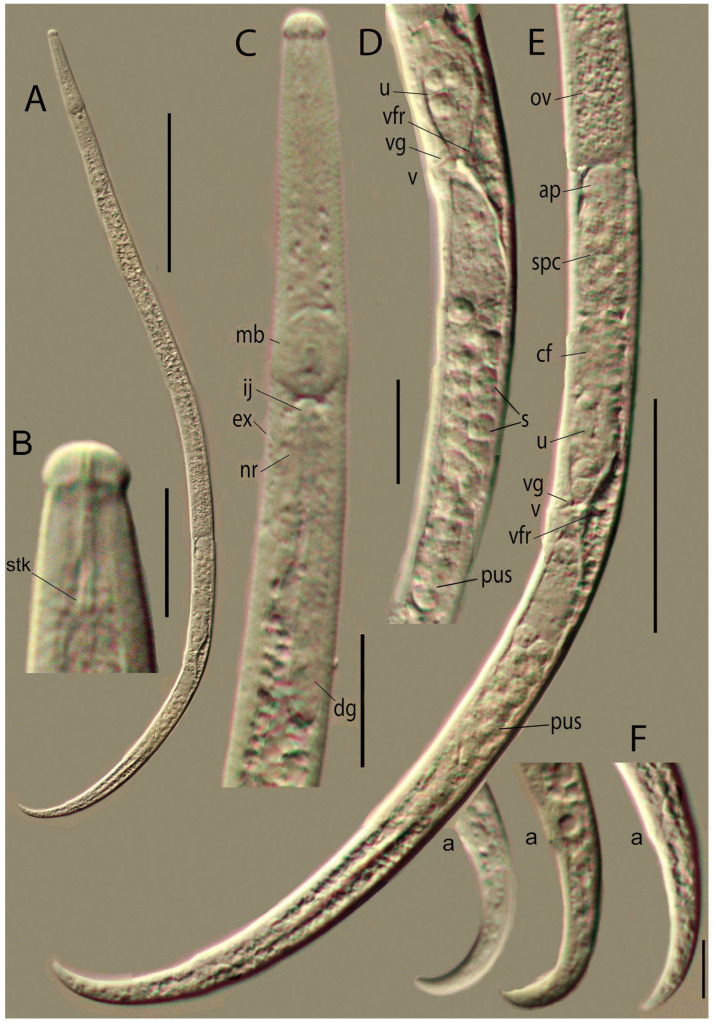
*Bursaphelenchus ussuriensis* sp. n. Female. Light microscopic photographs. (**A**)—total body outline. (**B**)—lip region. (**C**)—anterior part of the body. (**D**)—vulval region. (**E**)—posterior part of the body. (**F**)—tails. Abbreviations: a—anus, ap—apex of spermatheca, cf—crustaformeria, dg—dorsal gland, ex—secretory–excretory pore, ij—pharingo–intestinal valve, mb—median bulb, nr—nerve ring, ov—ovary, pus—postuteral sac, s—sperm, spc—spermatheca, stk—stylet knobs, u—uterus, v—vulva, vfr—vaginal framework, vg—vagina. Scale: 100 µm for (**A**); 10 µm for (**B**,**F**); 20 µm for (**C**,**D**); 50 µm for (**E**).

**Figure 4 plants-14-00093-f004:**
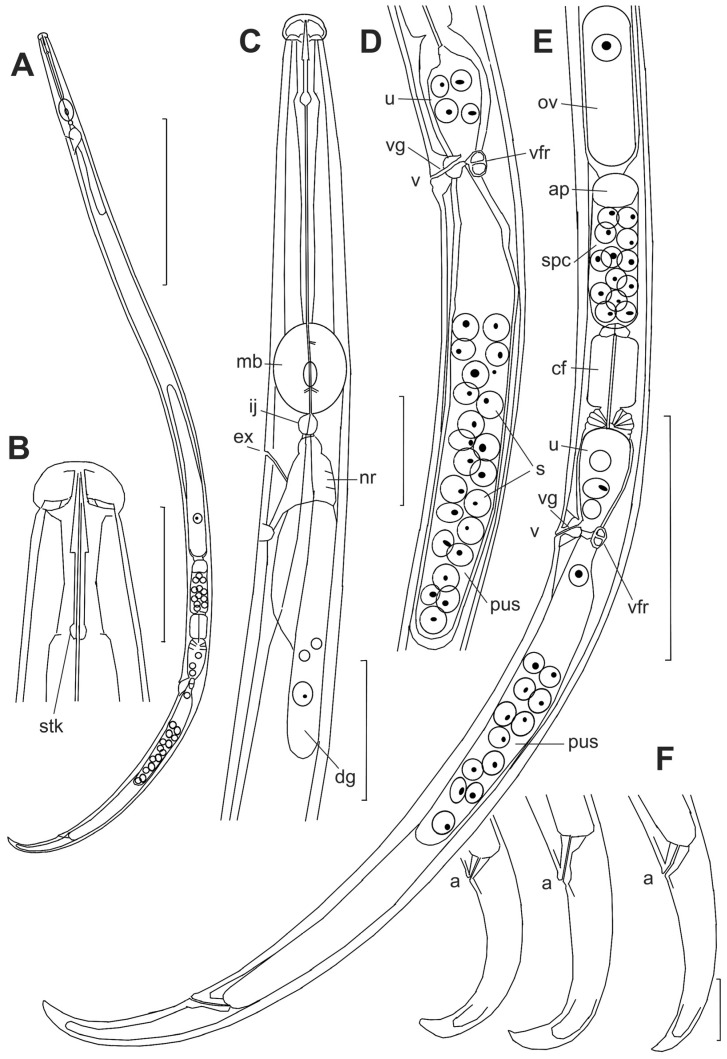
*Bursaphelenchus ussuriensis* sp. n. Drawings. Female. (**A**)—total body outline. (**B**)—lip region. (**C**)—anterior part of the body. (**D**)—vulval region. (**E**)—posterior part of the body. (**F**)—tails. Abbreviations: a—anus, ap—apex of spermatheca, cf—crustaformeria, dg—dorsal gland, ex—secretory–excretory pore, ij—pharingo–intestinal valve, mb—median bulb, nr—nerve ring, ov—ovary, pus—postuteral sac, s—sperm, spc—spermatheca, stk—stylet knobs, u—uterus, v—vulva, vfr—vaginal framework, vg—vagina. Scale: 100 µm for (**A**); 10 µm for (**B**,**F**); 20 µm for (**C**,**D**); 50 µm for (**E**).

**Figure 5 plants-14-00093-f005:**
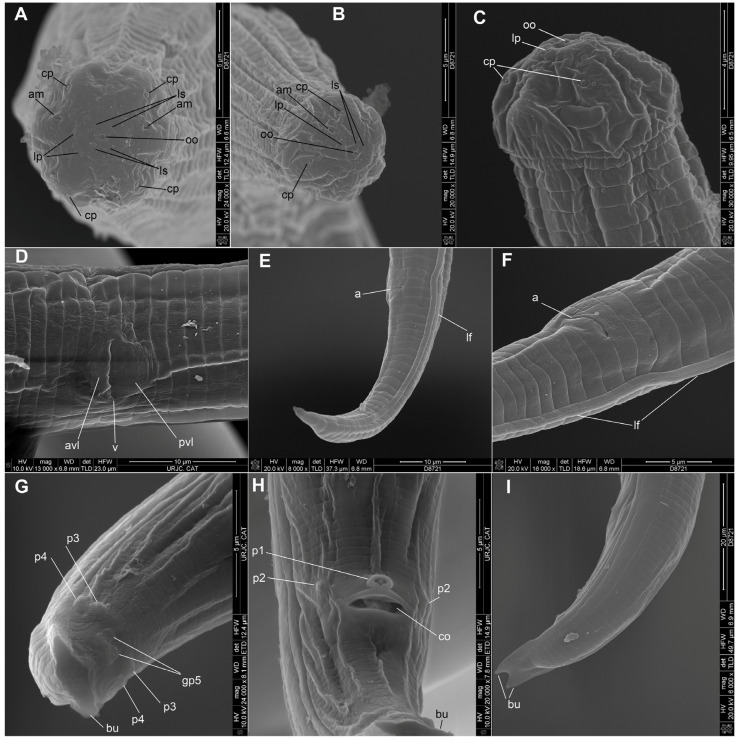
*Bursaphelenchus ussuriensis* sp. n. Scanning electron microscopic photographs. (**A**,**B**)—male lip region; am-amphid, cp—cephalic papillae, ls—inner lip sensillae, lp—outer lip papillae, oo—oral opening. (**C**)—female lip region. (**D**)—vulval region, avl—anterior vulval lip, pvl—posterior vulval lip, v—vulva. (**E**)—female tail. (**F**)—anal region. a—anus; lf—lateral field. (**G**–**I**)—male tail. bu—bursa, gp5—glandpapillae, p1—unpaired anterior papilla, p2—paired anterior papilla, p3, p4—paired tail papillae. Scales are indicated on the right or under each image.

**Figure 6 plants-14-00093-f006:**
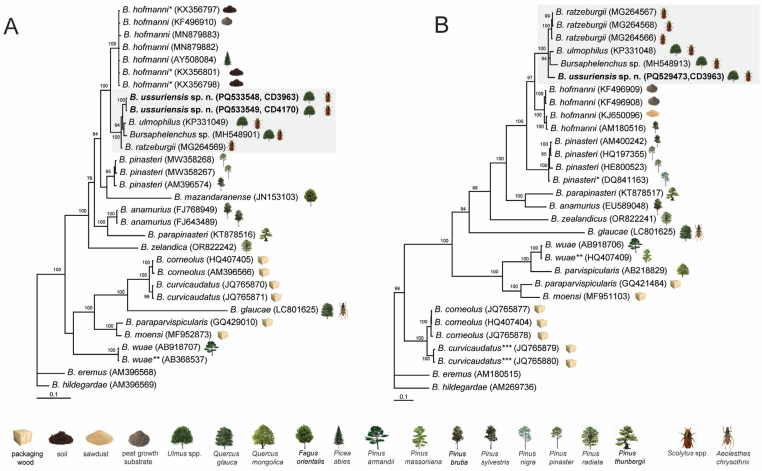
Phylogenetic relationships of *Bursaphelenchus ussuriensis* sp. n. within some species of the *Hofmanni* group *sensu* Ryss et al. [8], as inferred from Bayesian analysis of the D2-D3 expansion segments of 28S rRNA (ntax = 27; nchar = 786) (**A**) and ITS rRNA (ntax = 27; nchar = 1329) (**B**) gene sequences. Posterior probability values of more than 70% are given on appropriate clades. New sequences are indicated in bold. (**A**): *—identified as *Bursaphelenchus* sp. in the GenBank, **—identified as *B. parvispicularis* in the GenBank; (**B**): *—identified as *B hofmanni* in the GenBank; **—identified as *B. yuyaoensis* in the GenBank, ***—identified as *B. corneolus* in the GenBank. The *B. ratzeburgii* species complex is indicated by a gray area.

**Figure 7 plants-14-00093-f007:**
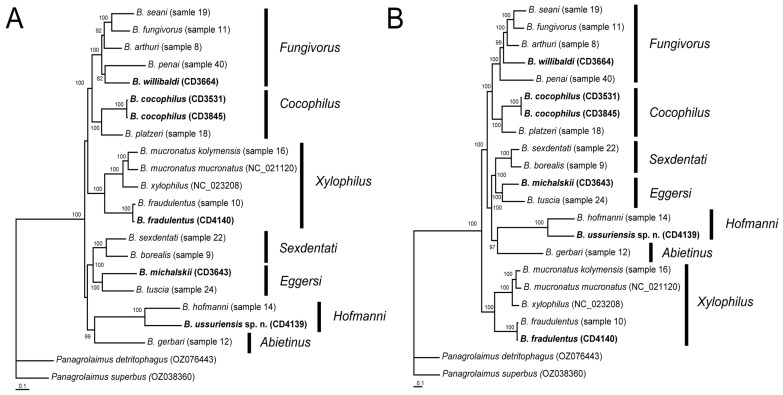
Phylogenetic relationships between *Bursaphelenchus* species as inferred from Bayesian analysis of the nucleotide sequences (ntax = 22; nchar = 10,471) (**A**) and amino acid sequences (ntax = 22; nchar = 3491) (**B**) of mitochondrial protein-coding genes. Posterior probability values of more than 70% are given on appropriate clades. New sequences are indicated in bold. *Bursaphelenchus* species groups are indicated. GenBank numbers of new sequences are given in Appendix A.

**Table 1 plants-14-00093-t001:** Morphometrics of *Bursaphelenchus ussuriensis* sp. n. All measurements are in µm and in the form of mean ± s.d. (range).

Sample Codes	CD3963 and CD4139			CD4140	
Character	Male		Female	Male	Female
	Holotype	Paratypes	Paratypes	Location-2	Location-2
n	1	20	20	20	20
L	660	605 ± 66 (497–698)	640 ± 76 (554–771)	648 ± 37 (596–700)	713 ± 30 (659–754)
a	45.8	44.3 ± 4.6 (38.1–53.7)	37.7 ± 1.4 (35.7–39.6)	47.9 ± 4.1 (41.6–53.8)	39.9 ± 1.9 (37.3–42.7)
b	11.2	10.5 ± 0.9 (9.1–11.8)	11.1 ± 1.2 (9.7–12.6)	11.0 ± 0.6 (10.1–12.1)	12.1 ± 0.8 (10.8–13.4)
b’	6.1	5.4 ± 0.7 (4.6–6.5)	5.4 ± 0.5 (4.5–6.3)	5.7 ± 0.5 (5.1–6.9)	6.2 ± 0.5 (5.7–6.8)
c	24,7	21.5 ± 1.5 (19.9–24.7)	17.6 ± 2.6 (14.4–23.1)	21.2 ± 1.6 (19.4–24.1)	16.9 ± 1.2 (15.1–19.0)
c’	2.2	2.5 ± 0.2 (2.2–2.9)	4.5 ± 0.4 (4.1–5.3)	2.6 ± 0.2 (2.2–3.0)	4.8 ± 0.5 (3.8–5.5)
T or V	72	64 ± 5 (56–72)	71 ± 2 (69–74)	57 ± 12 (31–74)	73 ± 1 (72–76)
Stylet	12	11.5 ± 0.9 (10.0–13.0)	11.9 ± 0.5 (11.0–12.5)	10.9 ± 1.3 (8.5–13.0)	11.0 ± 0.7 (10–12)
Stylet cone/stylet, %	45	50 ± 4 (45–56)	55 ± 6 (43–62)	46 ± 2 (42–50)	47 ± 3 (42–55)
Cephalic region diam.	6.5	6.2 ± 0.4 (5.8–6.6)	6.7 ± 0.4 (6.0–7.5)	6.1 ± 0.2 (6.0–6.5)	6.2 ± 0.3 (6.0–6.5)
Cephalic region height	3.0	2.9 ± 0.3 (2.5–3.5)	3.1 ± 0.4 (2.5–3.5)	2.7 ± 0.3 (2.0–3.5)	2.9 ± 0.2 (2.5–3.5)
Median bulb length (L)	14	13.3 ± 0.6 (12.5–14.0)	14.3 ± 0.9 (13.5–16.0)	13.6 ± 0.5 (13.0–14.5)	13.9 ± 0.8 (12.5–15.0)
Median bulb diam. (D)	10	9.3 ± 0.5 (8.5–10.0)	9.5 ± 0.7 (8.5–10.5)	9.4 ± 0.4 (9.0–10.0)	9.8 ± 0.7 (8.5–11.0)
Median bulb L/D	1.4	1.4 ± 0.1 (1.4–1.5)	1.5 ± 0.1 (1.4–1.6)	1.4 ± 0.1 (1.3–1.6)	1.4 ± 0.1 (1.3–1.5)
Median bulb valve	3	2.8 + 0.4 (2.2–3.0)	3.6 ± 0.5 (3.0–4.4)	3.5 ± 0.4 (3.0–4.0)	3.7 ± 0.4 (3.0–4.3)
Median bulb valve position/ Med. bulb. L, %	56	52 + 3 (48–56)	56 ± 3 (53–61)	55 ± 3 (51–61)	56 ± 2 (53–59)
Secretory–excretory pore from anterior	60.5	64 ± 5 (58–74)	62 ± 4 (56–68)	67 ± 4 (59–74)	69 ± 7 (58–82)
Hemizonid from anterior	63	73 ± 6 (63–80)	70 ± 5 (65–76)	70 ± 5 (62–80)	73 ± 6 (63–87)
Pharynx	59	57.6 ± 1.4 (55–59)	57.6 ± 2.2 (55–61)	58.8 ± 2.4 (55–63)	59 ± 5 (50–64)
Anterior to gland lobe end	109	113.3 ± 5.7 (107–123)	117.9 ± 10.5 (107–131)	114.0 ± 7.1 (100–130)	115.8 ± 10.7 (100–130)
Gland lobe	50	55 ± 6 (49–66)	60 ± 10 (49–74)	55 ± 7 (43–69)	56 ± 8 (46–66)
Gland lobe/ body diam. at median bulb.	3.7	4.5 ± 0.6 (3.6–5.3)	4.9 ± 0.8 (3.9–6.2)	4.3 ± 0.6 (3.1–5.4)	4.2 ± 0.5 (3.4–4.9)
Gland lobe/median bulb L	3.6	4.2 ± 0.5 (3.6–5.1)	4.2 ± 0.5 (3.6–5.0)	4.1 ± 0.5 (3.2–5.0)	4.1 ± 0.6 (3.2–4.9)
Max. body diam. (D)	14.5	13.7 ± 1.6 (11.5–16.0)	17.1 ± 2.5 (14.0–21.0)	13.6 ± 0.9 (12.5–13.5)	17.9 ± 0.9 (16.0–19.0)
Posterior genital branch	-	-	82 ± 11 (72–99)	-	82 ± 7 (68–90)
Posterior genital branch/vulval diam.	-	-	4.8 ± 0.6 (3.7–5.4)	-	4.6 ± 0.5 (3.6–5.3)
Posterior genital branch/vulva–anus distance (%)	-	-	56 ± 10 (39–66)	-	54 ± 5 (42–60)
Tail	26.0	28.3 ± 3.5 (23.5–34.5)	36.7 ± 4.7 (30.0–46.0)	30.7 ± 2.1 (27–35)	42.4 ± 2.9 (38–48)
Tail diam.	12.0	11.4 ± 0.8 (10.5–12.5)	8.1 ± 0.9 (7.0–10.0)	11.8 ± 0.5 (11–13)	9.0 ± 0.9 (8–11)
Annuli (width of 10 at mid-body)	12	10.8 ± 1.3 (9.0–12)	11.6 ± 1.1 (9.5–13.0)	10.3 ± 1.3 (9–12)	10.7 ± 1.9 (9.5–13.0)
Spicule length (arc, mid-line)	9.5	10.3 ± 0.9 (9.5–12.5)	-	11.3 ± 0.8 (10.0–12.5)	-
Spicule length (arc, dorsal)	13.5	13.2 ± 0.7 (12.0–14.0)	-	14.1 ± 0.8 (13.0–15.5)	-
Spicule length (arc, ventral)	9.0	9.0 ± 0.8 (7.5–10.0)	-	9.7 ± 0.7 (9–11)	-
Spicule arc/width posterior to rostrum (lateral view)	3.3	2.8 ± 0.4 (2.5–3.3)	-	3.4 ± 0.3 (3.0–4.0)	-
Ratio: distance from line between spicule rostrum and condylus ends to bottom of capitulum depression/rostrum-condylus length	0.24	0.18 ± 0.04 (0.15–0.25)	-	0.21 ± 0.06 (0.12–0.30)	-
Spicule length (along arc)/capitulum width (distance between ends of rostrum and condylus)	1.8	1.8 ± 0.2 (1.7–2.2)	-	2.0 ± 0.2 (1.6–2.2)	-
Angle between lines: along capitulum (condylus-rostrum) and extending spicule end (dorsal intersection)	−8.0	−7.5 ± 2.1 (−10.0–4.8)	-	−8.6 ± 1.6 (−11.2–6.3)	-

## Data Availability

New sequences of *B. ussuriensis* sp. n., *B. cocophilus, B. fraudulentus, B. michalskii*, and *B. willibaldi* were submitted to the NCBI database under accession numbers: PQ474985-PQ475026, PQ528008-PQ528012, PQ590176-PQ590199 (Appendix A) and PQ529473 (ITS rRNA gene), Q533548, PQ533549 (D2-D3 of 28S rRNA gene) and PQ500562 (18S rRNA-ITS1-5.8S rRNA-ITS2-28S rRNA gene) for *B. ussuriensis* sp. n. Nematode materials were deposited in the Nematode Collection of the Zoological Institute of the Russian Academy of Sciences (UFK ZIN RAS).

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
