# Peer review of "Wood-Inhabiting Nematode, Bursaphelenchus ussuriensis sp. n. (Nematoda: Aphelenchoididae) from David Elm, with Molecular Phylogeny of the Genus Based on Partial Mitochondrial Genomes"

_plants, 2024, doi:10.3390/plants14010093_

Round 1

Reviewer 1 Report

Comments and Suggestions for Authors

The manuscript is a description of a new species of a  plant parasitic nematode from the genus Bursaphelenchus and includes a phylogeny of the Hoffmanni group of the Bursaphelenchus genus based on mitochondrial genes that will be useful for establishing relationships within the group of these plant parasitic  nematodes. Several species of this genus are serious pests of woody plants.

Description, tables, figures and pictures are complete and detailed. Diagnosis of the new species is well established.

Description of adults was based on specimens obtained from in vitro cultures at the lab, with the inoculum of Dauer juveniles obtained from the dead wood. I wonder if a description of some adults obtained directly from the wood was not possible. Could it be possible that nematode culture in vitro had select individuals somehow morphologically different than those from the wild population?

The manuscript is acceptable in its present form or just with some minor edition. I have added just some minor suggestions in the pdf file.

Author Response

Comments: The manuscript is acceptable in its present form or just with some minor edition. I have added just some minor suggestions in the pdf file

Reply. Thanks! All referee corrections are introduced in the revised text of the manuscript.

Comments: Could it be possible that nematode culture in vitro had select individuals somehow morphologically different than those from the wild population?

Reply. Unfortunately, mainly dauer juveniles were collected and used for cultures. We were not also able to compare from adults from cultures and original probes.     

Reviewer 2 Report

Comments and Suggestions for Authors

In this manuscript, the authors describe the characteristics of a novel nematode, Bursaphelenchus ussuriensis, and provide a detailed comparison with previously reported nematodes. The findings significantly contribute to our understanding of the diversity within the Bursaphelenchus family. The data is well-organized, clearly presented, and appropriately discussed.

Minor points

1.     In line 94, the authors reference "Figures 1-5, Table 1." However, there is no further notation of the results after this point. I believe this approach could be improved. The authors should reference specific figures or tables as they describe the information, using notations such as "Fig. 1 A, Fig. 1 B," etc. This would help readers locate detailed features more easily.

2.     The authors discussed the most significant differences between *Bursaphelenchus ussuriensis* and other members of the Hofmanni group in lines 239-242. It would be beneficial for the authors to provide a supplementary figure that compares detailed features, such as the male caudal papillae (p3 and p4). This would aid readers who may not be well-versed in nematode taxonomy.

Author Response

Comment 1: In line 94, the authors reference "Figures 1-5, Table 1." However, there is no further notation of the results after this point. I believe this approach could be improved. The authors should reference specific figures or tables as they describe the information, using notations such as "Fig. 1 A, Fig. 1 B," etc. This would help readers locate detailed features more easily.

Reply: Thanks for this comment. We introduced specific references across the description as recommended.

Comment 2: The authors discussed the most significant differences between *Bursaphelenchus ussuriensis* and other members of the Hofmanni group in lines 239-242. It would be beneficial for the authors to provide a supplementary figure that compares detailed features, such as the male caudal papillae (p3 and p4). This would aid readers who may not be well-versed in nematode taxonomy.

Reply: Thanks for this comment. We added Fig. S1. Male papillae pattern in several species groups of Bursaphelenchus with explanations in Supplementary Materials.